# Global Dynamic Rainfall-Induced Landslide Susceptibility Mapping Using Machine Learning

Bohao Li [1], Kai Liu [1,2,*], Ming Wang [1], Qian He [1], Ziyu Jiang [1], Weihua Zhu [1] and Ningning Qiao [1]

1   School of National Safety and Emergency Management, Beijing Normal University, 19 Xinjiekou Wai Ave., Beijing 100875, China
2   Collaborative Innovation Center on Forecast and Evaluation of Meteorological Disasters (CIC-FEMD), Nanjing University of Information Science & Technology, Nanjing 210044, China
*   Correspondence: liukai@bnu.edu.cn

**Abstract:** Precipitation is the main factor that triggers landslides. Rainfall-induced landslide susceptibility mapping (LSM) is crucial for disaster prevention and disaster losses mitigation, though most studies are temporally ambiguous and on a regional scale. To better reveal landslide mechanisms and provide more accurate landslide susceptibility maps for landslide risk assessment and hazard prediction, developing a global dynamic LSM model is essential. In this study, we used Google Earth Engine (GEE) as the main data platform and applied three tree-based ensemble machine learning algorithms to construct global, dynamic rainfall-induced LSM models based on dynamic and static landslide influencing factors. The dynamic perspective is used in LSM: dynamic changes in landslide susceptibility can be identified on a daily scale. We note that Random Forest algorithm offers robust performance for accurate LSM (AUC = 0.975) and although the classification accuracy of LightGBM is the highest (AUC = 0.977), the results do not meet the sufficient conditions of a landslide susceptibility map. Combined with quantitative precipitation products, the proposed model can be used for the release of historical and predictive global dynamic landslide susceptibility information.

**Keywords:** dynamic landslide susceptibility; machine learning; rainfall; global scale

## 1. Introduction

Landslides are one of the many natural hazards that are responsible for mass destruction and loss of life worldwide. Landslides can occur following a variety of highly complex mechanisms; most events are triggered by extreme precipitation, snowmelt, earthquakes, water level changes, erosion and weathering, and human activities, but prolonged heavy precipitation on saturated slopes is the main mechanism [1,2]. In addition to inflicting serious economic damage, destroying infrastructure, and affecting landscape and river ecosystems, landslides can cause mortality on a large scale, with a total of 55,997 people having lost their lives in the 4862 fatal landslides recorded globally between 2004 and 2016 [3]. In addition, the distribution of landslides in space is heterogeneous, and they have particularly significant impacts on high-incidence areas [3]. Therefore, research on the intensity and spatial distribution of rainfall-induced landslide risk is of considerable importance for disaster loss reduction and emergency management planning endeavours worldwide [4]. A key step in assessing landslide risk is landslide susceptibility mapping (LSM), whose mapping results can reveal the spatial distribution of the future landslide likelihood and provide decision-making support to planners regarding future development [5–7].

Machine learning algorithms are increasingly being used in LSM because they can effectively handle the linear or nonlinear relationships between landslide impact factors and landslide occurrence at different scales and from different sources [8,9]. The more common machine learning algorithms applied to LSM in recent years include logistic regression (LR) [10,11], maximum entropy (MaxEnt) [12–14], artificial neural networks (ANNs) [15,16], support vector machines (SVM) [17,18], random forest (RF)

algorithms [6,19], k-nearest neighbour (KNN) clustering [20], decision trees (DTs) [6,21], and gradient boosting (GB) [22,23]. Accordingly, many studies have sought to compare the performance of different algorithms in LSM. For instance, taking a region in Saudi Arabia as the study area, Youssef and Pourghasemi [24] compared seven different machine learning algorithms (including SVM, RF, ANNs, and linear discriminant analysis (LDA)) and found that RF and LDA achieved the best relative model performance. Through a review of the literature, Merghadi et al. [25] identified most of the machine learning algorithms employed in mainstream LSM research in recent years and compared their modelling accuracies. They found that all tree-based ensemble models—RF, extremely randomized trees (ET), and light gradient boosting machine (LGB)—performed better than other machine learning models. Huang and Zhao [26] discussed the advantages and disadvantages of the SVM algorithm in LSM in comparison with LR, ANN, and RF and reported that SVM did not always achieve the best accuracy but that it can more easily solve nonlinear and high-dimensional classification problems. MaxEnt is a statistical probabilistic machine learning algorithm which is often used in various hazard susceptibility assessments [27–29]. It also has been used in LSM to construct hybrid models with other machine learning algorithms to improve performance [30–32]. However, no specific model has been shown to be suitable for all LSM scenarios. In addition, models that perform better in binary classification are not necessarily more suitable for LSM, since the LSM result is related to the probability of the model determining positive and negative samples. Ultimately, the selection of machine learning algorithms for LSM should take the combination of performance and time efficiency into account [33].

LSM has been the subject of research for many years, but most were studied at a regional scale. In fact, the global LSM is meaningful for several reasons: it is useful in areas where landslide data are scarce, can help researchers to identify patterns in the spatial distribution of landslide occurrence in different environments, and can provide a reference benchmark for different regions to communicate and compare their disaster prevention and mitigation strategies [34–38]. Moreover, global landslide susceptibility maps have been applied to landslide risk assessment [39–41] as well as to global landslide early warning systems [42,43]. Some global-scale LSM studies have achieved reasonable assessment results [2,44,45], but it is difficult to overcome the spatial heterogeneity of landslides, the difficulty of selecting landslide influencing factors, and the limited spatial resolutions of related datasets. Among these issues, the selection of landslide influencing factors has a particularly significant impact on the modelling accuracy. Dai and Lee [46] proposed that the factors leading to landslides can be divided into dynamic and static factors. Dynamic factors, mainly precipitation and seismic activity, are capable of directly triggering large-scale movements of the earth's crust. Nevertheless, research on landslide susceptibility is based predominantly on geological, geomorphological, soil, environmental, and other related factors without considering the influences of dynamic changes in related factors. However, there are many dynamic factors in addition to precipitation and earthquake, for instance, the surface cover changes with obvious seasonality can also affect the occurrence of landslides. More importantly, most studies that considered the effects of rainfall chose the average precipitation over a period and held all other static factors constant. In contrast, for a landslide triggered by precipitation, only the precipitation days before the landslide should be directly related to the landslide occurrence. So, broadening the scope of LSM, current hazard susceptibility evaluations are mostly conducted to reflect the multi-year average hazard susceptibility. This approach, known as dynamic landslide susceptibility mapping [47], is performed by extracting the influencing factors of landslides corresponding to historical landslide occurrence times and locations, constructing a model, and assessing dynamic landslide susceptibility in conjunction with dynamic variables such as precipitation data. The results of dynamic LSM can reflect the spatial and temporal variabilities of landslide susceptibility. Dynamic landslide susceptibility analysis can reveal more about the relationship between landslide occurrence and landslide influencing factors, and can help researchers in related fields to understand the mechanism of landslides more

accurately. In addition, the dynamic landslide susceptibility maps are more suitable for risk analysis and early warning system development with its clear time scale and high accuracy. With the development of multi-source remote sensing technology, a growing number of landslide influencing factors can expose this variability at different times, thereby improving the LSM accuracy.

However, when constructing a landslide susceptibility model on a global scale, dynamic LSM is burdened by an overabundance of data regarding the dynamic factors of all landslide events. Fortunately, considerable technological advances have been achieved in cloud computing in recent years. Thus, the use of the Google Earth Engine (GEE) Python API with geemap zonal statistics and local download tools could be a good solution to this problem. GEE integrates multisource remote sensing data with other types of products and provides powerful data preprocessing functionality, allowing users to process data more quickly.

By deviating from the most static traditional research on LSM, this study adopts a dynamic and analytical perspective based on machine learning. In this study, we assess rainfall-induced landslide susceptibility using three machine learning models (RF, ET, and LGB) in combination of GEE and considering daily dynamic change of LSM. Then, based on the model performance and time efficiency, the best-performing algorithm in the dynamic LSM is selected to construct a high-precision, global, dynamic model for rainfall-induced landslide susceptibility mapping.

## 2. Materials

### 2.1. Landslide Inventory

The landslide data used in this study were obtained from the Cooperative Open Online Landslide Repository (COOLR). This website shares landslide point data from 1915 to June 2021, with each point containing detailed information on the hazard data sources, occurrence time, type, size, and coordinates (longitude and latitude), among other parameters, all of which are crucial for landslide hazard research. The major sources of information were online news articles and the public, and the landslide points had high spatial and temporal accuracies. COOLR is mainly made up of Global Landslide Catalog (GLC) [48,49], Landslide Reporter Catalog (LRC) [50], and some locally reported landslides. Most global LSM studies are based on data in COOLR. For example, Anne Felsberg et al. [51] used 12,515 hydrologically triggered landslide events from COOLR to analyse uncertainties in global LSM; Lin et al. [45] created a global landslide susceptibility map based on GLC and the World Geological Hazard Inventory; Kirschbaum and Stanley [44] studied the global susceptibility of landslide triggered mainly by rainfall based on 1194 rainfall-induced landslide events in the GLC and eight regional landslide databases. Relevant data sources and references in COOLR are shown in Table S1. After integrating all the data, we extracted only the landslide events that were triggered by precipitation between 2000 and 2020, yielding a total of 9223 events. Figure 1 shows the spatial distribution of landslides triggered by all mechanisms in this dataset.

### 2.2. Influencing Factors

The selection of appropriate influencing factors is a critical step for LSM, as it can affect the predictive competences of numerous models [52]. We summarised the relevant literature and found that precipitation, geological features, soil features, geomorphometric features, and environmental features are the most widely used landslide influencing factors in relevant studies worldwide [45,53–58]. In this study, we selected 17 influencing factors that may affect landslide susceptibility, as shown in Table 1. They are divided into dynamic and static influencing factors, with dynamic influencing factors based on time-series data, including normalised difference vegetation index (NDVI), land cover, and precipitation. Data regarding precipitation, soil, terrain, NDVI, and land cover corresponding to the landslide points were extracted based on the GEE platform. Other influencing factors were derived from various data sources, as given in Table 1. The information of these features is visualised in Figures S1 and S2.

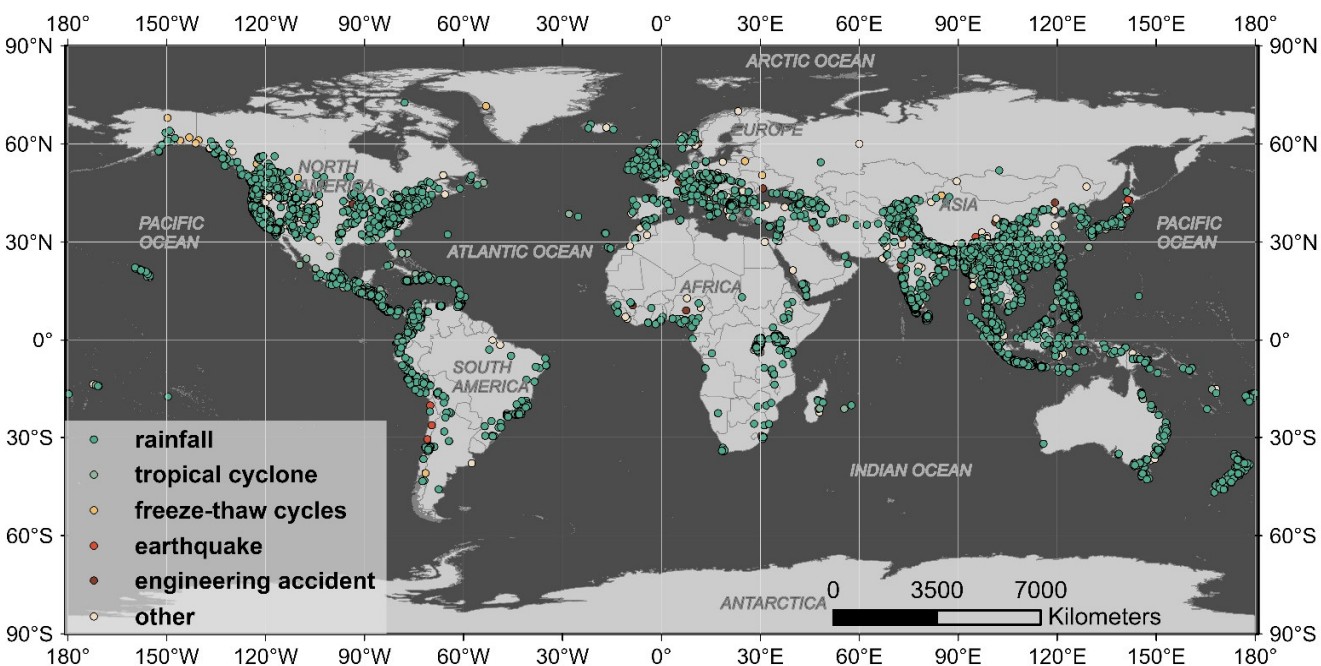

**Figure 1.** Spatial distribution of landslide events with different triggering factors. The landslide data source is COOLR.

### 2.2.1. Geological Features

In particular, lithology directly affects the stability of slopes and is a very important landslide influencing factor [59]. The geological data were extracted from the Global Lithological Map (GLIM), Global Active Faults Database (GAF-DB), and Global Seismic Hazard Map. The first-level GLIM dataset contains 16 categories, including evaporites, ice and glaciers, metamorphics, etc. [60]. The raw GLIM GIS data were downloaded from the website of the Commission for the Geological Map of the World (CCGM.ORG) and converted into raster data at a spatial resolution of 250 m. Because faults are also a critical factor affecting slope stability [61], we further extracted the distances from landslide points to faults. The fault data were downloaded from the Global Earthquake Model (GEM) Foundation GAF-DB published in 2019 (https://github.com/GEMScienceTools/gem-global-active-faults/, accessed on 3 January 2022), which contains data on 13,500 faults worldwide with good spatial integrity [62]. As another major cause of landslides, earthquakes sometimes easily act in tandem with precipitation to trigger landslides, therefore, the return period of the peak ground acceleration (PGA) should be an influencing factor of rainfall-induced landslides. Seismic data were downloaded from the Global Seismic Hazard Map (https://hazard.openquake.org/gem/, accessed on 21 December 2021), which shows the global spatial distribution of the PGA with a return period of 475 years at a spatial resolution of 0.04° [63].

### 2.2.2. Soil Feature

Generally, all kinds of soil may form landslides, but unconsolidated soil with poor weathering resistance that easily deforms due to precipitation is most likely to cause landslides [64,65]. OpenLandMap USDA Soil Taxonomy Great Groups were used in this study and the data were sourced from GEE. The data can be accessed and visualised at OpenLandMap.org.

### 2.2.3. Geomorphometric Features

Geomorphometric features include elevation, slope, the terrain ruggedness index (TRI), the topographic wetness index (TWI), and curvature. Topography controls the flow of water and the concentration of soil moisture, so it is a vital landslide influencing factor [10]. The

digital elevation model (DEM) data was from NASA's Shuttle Radar Topography Mission (SRTM) Digital Elevation Database Version 4 [66]. Since SRTM DEM is not available at high latitudes, the Global Multiresolution Terrain Elevation Data 2010 (GMTED2010) model with a resolution of 7.5 arc seconds (approximately 230 m) were used to fill this gap [67]. The Root Mean Square Error (RMSE) for these two products is approximately 9.73 m and 26–30 m. The STRM DEM and GMTEDTED2010 were used to derive terrain factors, namely, elevation, slope, TRI, TWI, curvature, plan curvature, and profile curvature.

### 2.2.4. Environmental Features

The surface cover is related to the infiltration conditions of surface runoff, which indirectly affects the occurrence of landslides. NDVI, as a measure of vegetation cover, is often used in LSM analysis [68]. In addition, human activities play a key role in modifying the environment, which have the potential to change vegetation cover conditions and alter the topography. For example, road construction activities may result in improper cut slopes and road drainage, which may cause landslides [69–71]. We consider NDVI, land cover, and distance to road as environmental features. NDVI and land cover are obtained from MODIS products (MOD13Q1 V6 and MCD12Q1 V6, respectively). The average monthly NDVI and annual land cover data were extracted for each landslide event according to the month and year when the landslide occurred, respectively. Distance to road is calculated as the Euclidean distance from the centre of each cell in the global 500 m grid to the world's major roads in the Pseudo-Mercator projected coordinate system, and the road data were downloaded from www.openstreetmap.org, accessed on 3 January 2022.

### 2.2.5. Precipitation Features

The relationship between the short-time precipitation before landslide and landslide occurrence is very strong [72–74]. For precipitation data, we used the ERA5-Land Hourly-ECMWF Climate Reanalysis dataset [75], which provides land climate variables from 1981 to 2022 with a spatial resolution of $0.1°$ and completely covers the global land area. The temporal resolution of the dataset is 1 h. To improve the modelling accuracy as much as possible, we selected precipitation-related variables, including precipitation on the day of landslide occurrence (Pr1d), cumulative precipitation 3 days before landslide occurrence (including precipitation on that day) (Pr3d), and cumulative precipitation 7 days before landslide occurrence (including precipitation on that day) (Pr7d). Similar to NDVI and land cover, we extracted these precipitation features from the time-series data for each event.

**Table 1.** Datasets and corresponding landslide influencing factors. The symbol "-" indicates that the data has no time-series information.

| Data Type | Dataset | Dataset Available Period | Resolution | Influencing Factor |
|---|---|---|---|---|
| Geological features | GLIM GEM GAF-DB | - - | 1:3,750,000 vectors | lithology distance to fault |
| | Global Seismic Hazard Map | - | $0.04°$ | 475-year return period PGA |
| Geomorphometric features | SRTM Digital Elevation Data Version 4 | - | 90 m | elevation slope TRI |
| | GMTED2010 | - | 7.5 arc-seconds | TWI curvature profile curvature plan curvature |
| Soil feature | OpenLandMap USDA Soil Taxonomy Great Groups | - | 250 m | surface soil taxonomy |
| Environmental features | MCD12Q1.006 MODIS Annual land cover type | 2001–2020 | 500 m | land cover |
| | OpenStreetMap global primary roads | - | vector | distance to road |
| | MOD13Q1.006 Terra Vegetation Indices 16-Day Global 250 m | 2000–2022 | 250 m | NDVI |
| Precipitation features | ERA5-Land Hourly-ECMWF Climate Reanalysis | 1981–2022 | $0.1°$ | Pr1d, Pr3d, Pr7d |

### 3. Methodology

The construction of a global dynamic LSM model using machine learning algorithms consists of the following steps: (1) generating a feature matrix by extracting dynamic and static landslide influencing factors based on historical landslide events, (2) selecting the major influencing factors for each landslide, (3) constructing models using machine learning algorithms, (4) evaluating the model performance, (5) selecting the mapping date and creating a fishnet to extract the main landslide influencing factors on the day of landslide occurrence, (6) producing a global, dynamic landslide susceptibility map at a resolution of 500 m, (7) analysing the sufficiency of the mapping result. The framework of the study is graphically depicted in Figure 2.

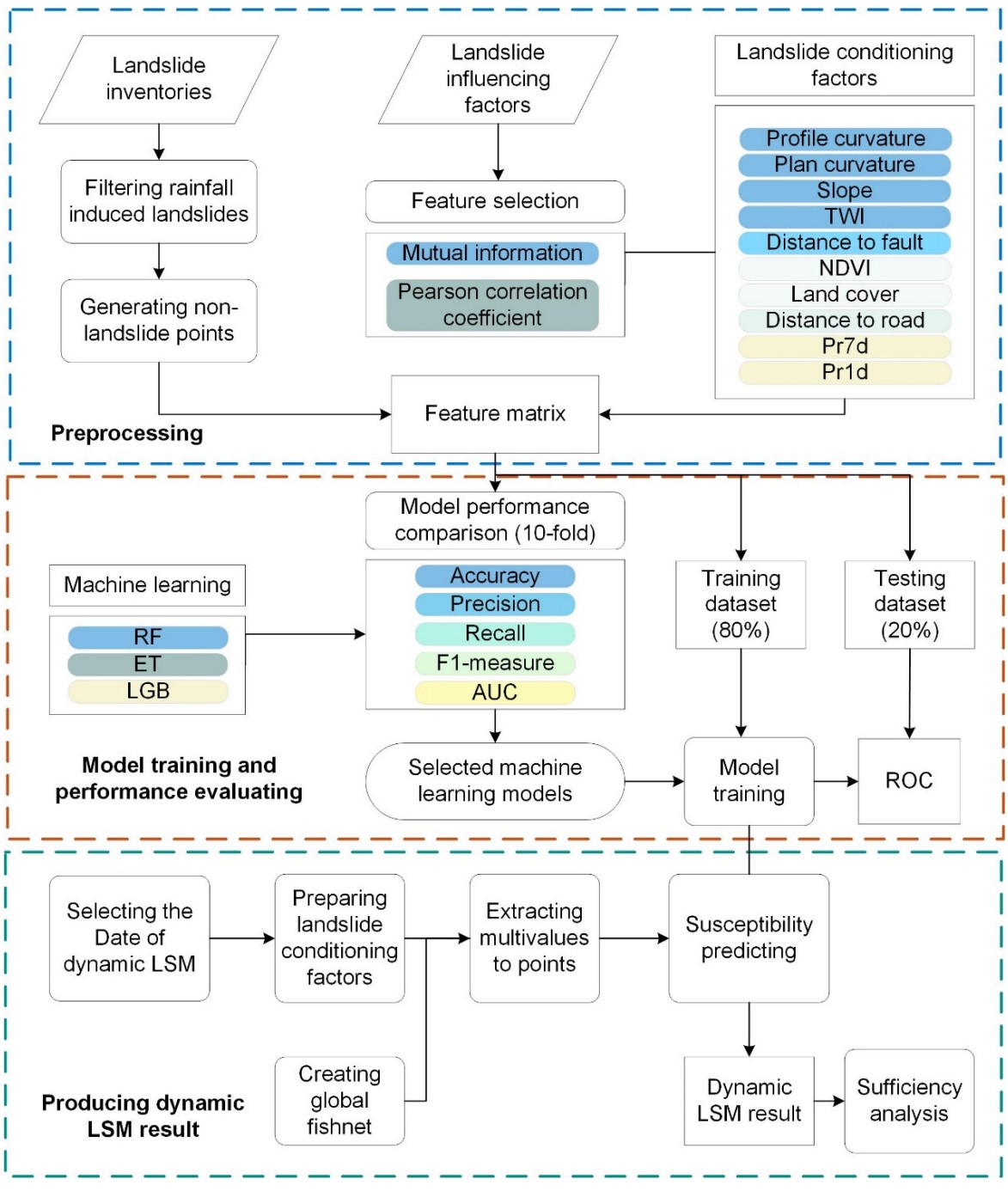

**Figure 2.** Workflow for constructing the global dynamic LSM model and comparing the performance of different models.

### 3.1. Machine Learning Algorithms

We compared three models that have achieved good LSM performance: RF, ET, and LGB. In this study, RF and ET are implemented using Python sklearn package, and LGB is based on lightgbm Library.

RF is a variant of the bagging method [76,77] based on DTs. DTs perform classification based on all features, while RF follows a different classifying approach. After randomly choosing the samples, RF further introduces random feature selection. For each node of the base learner, a subset containing k features is first selected from the feature set of the node, and then an optimal feature is selected from the subset for division. Here, k introduces randomness into the model. The core idea of RF is to construct multiple independent estimators and then determine the prediction result via either averaging or majority voting.

The ET method is a variant of the RF algorithm [78]. ET differs from RF in that each DT does not use a random sampling approach but directly uses the original training dataset. After each node of the base learner selects a subset of features, a feature is randomly selected from this subset for classification. Compared with RF, ET tends to result in larger DTs, and the variance of ET is further reduced, but the bias is further increased. In certain cases, the generalisation ability of ET is better than that of RF.

LGB, sometimes abbreviated LightGBM, is an emerging gradient boosting decision tree (GBDT) algorithm proposed by Microsoft in 2017, mainly with the goal of solving the problems encountered by GBDT when faced with massive data. LGB is a histogram-based DT algorithm that uses two new techniques, namely, gradient-based one-side sampling (GOSS) and exclusive feature bundling (EFB), to accelerate the model. Moreover, a leafwise tree growth strategy is used to improve the accuracy of the model and to avoid overfitting by limiting the tree depth. LGB supports efficient parallel computing with a small memory footprint [79] and improves the speed of conventional GBDT training by more than 20 times while achieving almost the same accuracy [80].

### 3.2. Generation of a Feature Matrix

The assessment of landslide susceptibility using machine learning algorithms is actually a binary classification issue. A certain number of non-landslide points are generated, and each point needs to have temporal attributes. The ensemble tree-based algorithms can perform well in the classification based on unbalanced samples [81–83]. We set the positive-to-negative sample ratio to 1:2 (where positive and negative samples represent landslide and non-landslide points, respectively) [84,85]. A total of 9223 records of rainfall-induced landslide events were available, so we randomly generated 18,446 non-landslide samples and ensured that these samples did not overlap in time and space.

In this study, dynamic variables were extracted to both landslide and non-landslide samples in GEE using geemap's point value extraction function. Static variables (soil features, elevation, and slope) were also directly extracted from GEE, whereas other features were extracted after being projected onto the WGS84 datum by ArcGIS Pro 2.8. Finally, all data were integrated to form a feature matrix.

### 3.3. Feature Selection

Dimensionality reduction is a preprocessing step that must be performed before implementing high-dimensional data analysis, visualisation, and modelling, and the easiest way to reduce dimensionality is feature selection [86]. This study adopted the following two steps to reduce the number of features while guaranteeing the modelling accuracy.

1. Mutual information

   The mutual information approach was employed to capture the linear and nonlinear relationships between each feature and class label. The mutual information between landslide influencing factor $X$ and landslide occurrence $Y$ is calculated as follows:

   $$I(X;Y) = \sum_{x \in X} \sum_{y \in Y} p(x,y) \log \frac{p(x,y)}{p(x)p(y)} \tag{1}$$

2. Pearson correlation coefficient

   In statistics, multicollinearity occurs when more than two explanatory variables are deeply correlated and the Pearson correlation coefficient is often used to quantify multi-collinearity [7,87]. The formula for the Pearson correlation coefficient is as follows:

   $$r = \frac{1}{n-1} \sum_{i=1}^{n} \left( \frac{X_i - \overline{X}}{\sigma_x} \right) \left( \frac{Y_i - \overline{Y}}{\sigma_y} \right) \tag{2}$$

   where r represents the correlation coefficient between factor $X$ and $Y$, n indicates the total number of samples, and $X_i$ and $Y_i$ represent the samples of factor $X$ and $Y$ indexed as i, respectively. $\overline{X}$ and $\overline{Y}$ refer to the sample means of factor $X$ and $Y$. $\sigma_x$ and $\sigma_y$ are sample standard deviations. r is between $-1$ and 1 and the linear relationship between two influencing factors increases with the growth of the absolute value of r. In this study, we determined that two influencing factors had a strong linear relationship when r is larger than 0.7 and the relationship was considered significant only when the *p*-value is less than 0.05 [88,89].

*3.4. Model Training and Performance Evaluation*

Evaluating the model performance is a necessary part of constructing a machine learning model, and a variety of different metrics are needed to evaluate the performance. For binary classification, the best evaluation metrics are defined based on the confusion matrix [90], including the overall accuracy (ACC), precision, recall, F1-measure, receiver operating characteristic (ROC) curve, and area under the ROC curve (AUC). The closer these metrics are to 1, the better the performance of the model. ROC is a curve with the false positive rate (FPR) as the horizontal axis and recall as the vertical axis at different thresholds. AUC is widely used to evaluate the LSM performance with the following criteria: excellent (0.9–1), very good (0.8–0.9), good (0.7–0.8), average (0.6–0.7), and poor (0.5–0.6) [25]. Table 2 lists the calculation methods and the focus of each performance evaluation metric.

**Table 2.** Performance evaluation metrics used to assess the binary machine learning model.

| Metric | Formula | Evaluation Focus |
| --- | --- | --- |
| ACC | $\frac{TP+TN}{TP+TN+FP+FN}$ | The proportion of landslide sample points which are correctly classified among all sample points |
| Precision | $\frac{TP}{TP+FP}$ | The ability to not classify non-landslide points as landslide points |
| Recall | $\frac{TP}{TP+FN}$ | The ability to correctly classify landslide points |
| F1-measure | $\frac{2TP}{2TP+FP+FN}$ | The ability to minimise the misclassification of non-landslide points when trying to correctly classify landslide points |
| FPR | $\frac{FP}{FP+TN}$ | The number of non-landslide points predicted as positive samples |
| AUC | Integral over the ROC curve. | The classification ability of landslide points and non-landslide points is considered together, which can overcome the sample imbalance problem |

Note: true positive (TP) and true negative (TN) denote the numbers of landslide and non-landslide points that are correctly classified, respectively. Likewise, false positive (FP) and false negative (FN) denote the numbers of misclassified landslide and non-landslide samples, respectively.

To compare the performance of the models, we set the number of trees to 300 for all three models and changed the boosting type of lightGBM to "goss". All other hyperparameters were used as default. The modelling accuracies and time efficiency of the different models were then compared in 10-fold cross validation. Moreover, we split the feature matrix into 80% training dataset for model training and 20% testing data for model evaluation.

### 3.5. Producing Dynamic Landslide Susceptibility Maps

Dynamic landslide susceptibility maps are time-dependent. Therefore, in this study, we randomly took two days, 15 December 2020 (Northern Hemisphere winter) and 15 June 2021 (Northern Hemisphere summer), as examples. After creating a global land fishnet consisting of 250 m blocks, the landslide influencing factors on these two dates were extracted. Then, we applied the model to assess the landslide susceptibility and rasterise the fishnet to produce two landslide susceptibility probability maps for these dates at a spatial resolution of 500 m. Finally, the landslide susceptibility probability map was divided into five categories by using the equal interval classification method with class names (probability ranges) of very low (0–0.2), low (0.2–0.4), moderate (0.4–0.6), high (0.6–0.8) and very high (0.8–1.0) [88,91,92].

To verify the reasonableness of the dynamic LSM results, we analysed their sufficiency. Generally, all the generated LSM results must fulfil two basic spatial principles: (1) the landslide density ratio increases from low-susceptibility classes to high-susceptibility classes, and (2) higher-susceptibility classes cover smaller areas [25,93,94]. The landslide density can be defined as the ratio of the number of landslide points falling in different categories to the area of the corresponding category [95]. Here, we projected the landslide point data and dynamic LSM results onto a cylindrical equal area map in ArcGIS Pro and quantified the area proportion and landslide density of each susceptibility category.

## 4. Results

### 4.1. Selection of Conditioning Factors

We sorted the mutual information between all the features and all the class labels and kept as few features as possible by using a learning curve while ensuring a high modelling accuracy of RF with 10 estimators in five-fold cross validation. The number of features after filtering using the mutual information approach was 13. They are distance to fault, slope, TRI, TWI, curvature, profile curvature, plan curvature, land cover, distance to road, NDVI, Pr1d, Pr3d, and Pr7d.

Figure 3 shows the Pearson correlation coefficient matrix of continuous influencing factors, and we removed TRI, curvature Pr3d to eliminate multicollinearity. The final landslide conditioning factors we finally retained were land cover, distance to fault, slope, TWI, profile curvature, plan curvature, distance to road, NDVI, Pr1d, and Pr7d.

### 4.2. Model Performance Assessment and Comparison

In terms of the model performance metrics, all three machine learning models perform well (AUC > 0.9), and the differences in each metric are not significant, as shown in Figure 4a. LGB had the highest modelling performance, with all metrics reaching their corresponding maximum among the three models. The average values of its five main performance metrics (accuracy, precision, recall, F1-measure, and AUC) reached 0.926, 0.888, 0.892, 0.890, and 0.977, respectively. The difference between RF and LGB was quite small, with the mean values of the five main performance metrics of RF decreasing by 0.09%, 0%, 0.25%, 0.01%, 0.13%, and 0.13% compared with those of LGB, and the modelling performance of ET is the poorest, with the ET metrics being lower than those of LGB by 0.36%, −0.24%, 0.27%, 0.68%, and 0.14%, respectively. Nevertheless, by comparing the standard deviations of these metrics, we found that LGB, RF, and ET exhibited a stable performance on different combinations of testing and training datasets. The maximum value of the standard deviation of all indicators for the three models was only 0.011. We

also counted the time of cross validation between the different models. LGB was the fastest, followed by ET and then by RF. Specifically, LGB was approximately three times faster than ET and five times faster than RF. Finally, we trained the three models on the training dataset for subsequent dynamic LSM, and Figure 4b shows the ROC curves for the three algorithms modelling on the testing dataset. LGB, RF, and ET all showed very good performance.

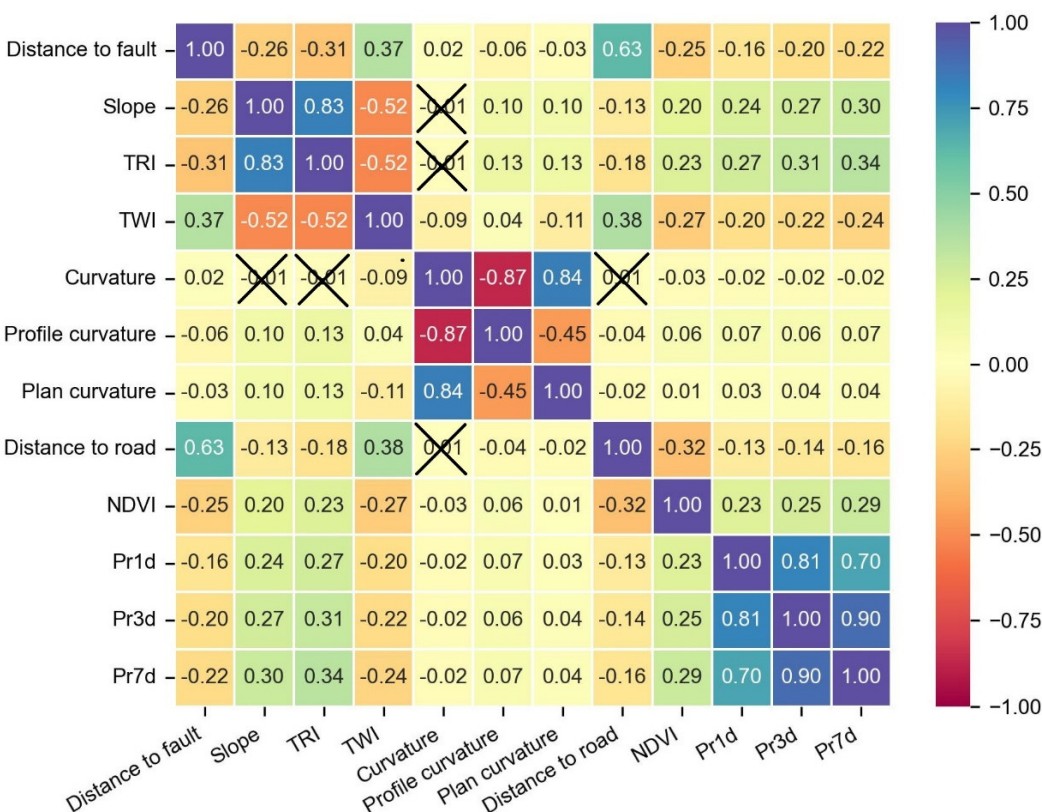

**Figure 3.** The Pearson correlation coefficient matrix of influencing factors (× represents the linear relationship is not significant).

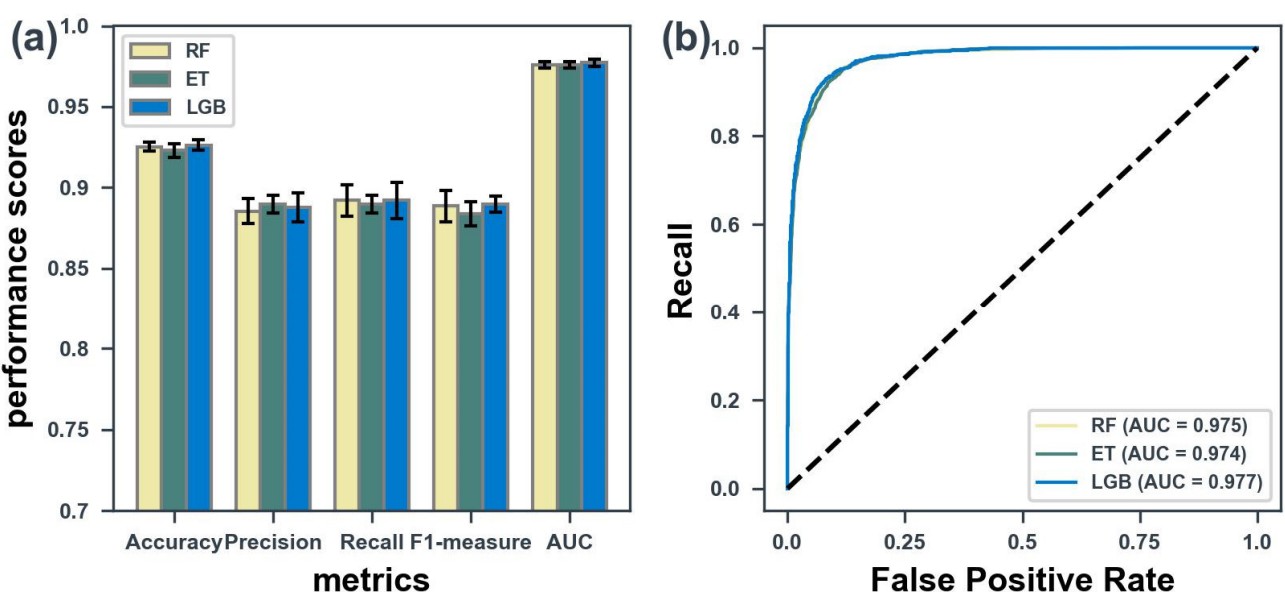

**Figure 4.** (**a**) Performance comparison among the three machine learning algorithms with 10-fold cross validation (**b**) Modelling ROC curves of the different machine learning algorithms.

### 4.3. Comparison of the Dynamic LSM Results on Two Sample Days

A comparison of the spatial distributions of the dynamic LSM results yielded by the three models generally reveals only minor differences (Figure 5). However, in some regions, such as Yunnan Province in Southwest China on 15 December 2020 (winter in the Northern Hemisphere with low rainfall), when the average Pr1d and Pr7d were only 0.5 mm and 2.9 mm, respectively, LGB returned a higher susceptibility assessment. In contrast, the results of RF and ET were quite different and were more consistent with the spatial distribution of precipitation (Figure S2). In addition, from a visual perspective, LGB seemed to yield more extreme assessments, and its binary classification probability predictions showed highly significant polarisation. In contrast, the probability prediction results of RF and ET are generally smooth.

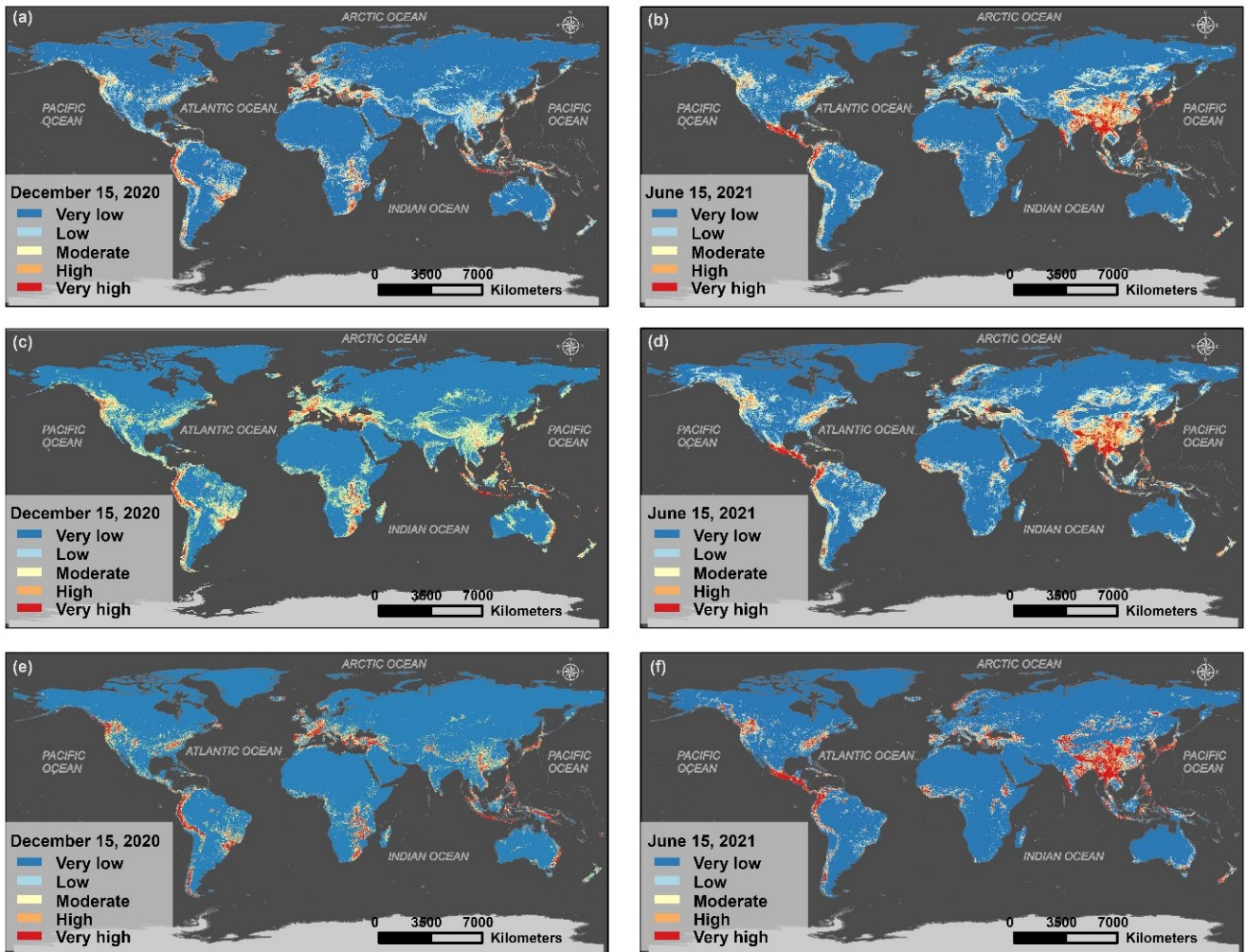

**Figure 5.** Dynamic LSM results of RF, ET, and LGB on the two example dates. Note: (**a**,**b**) are the mapping results of the RF model. (**c**,**d**) are the results of the ET model. (**e**,**f**) are the results of the LGB model.

Table 3 shows the statistical information for the above susceptibility maps. The analysis of their sufficiency is as follows: (1) The assessment results of RF, ET, and LGB satisfy the condition that the landslide density gradually increases from low- to high-susceptibility areas. (2) The area proportions of the very low and very high categories in the LGB assessment are significantly higher than those in the RF and ET assessments, which confirms that LGB yields binary classification predictions with greater confidence and gives more extreme probabilities. However, the landslide densities predicted by LGB in the high and very high categories are lower than those predicted by both RF and ET, indicating

that the assessment results of RF and ET could better match the spatial distribution of historical landslide points. (3) In the LGB assessment results, the proportion of the global area occupied by the very high category is apparently higher than that occupied by the moderate and high categories, which does not satisfy the second requirement for landslide susceptibility maps, i.e., that higher-susceptibility classes cover smaller areas. In contrast, the assessment results of RF and ET pass this test.

Ultimately, RF achieved a high modelling performance and exhibited a higher time-efficiency, which can meet the needs of dynamic LSM. Thus, the RF algorithm was selected to construct the global, dynamic rainfall-induced LSM model. We divided each precipitation feature into five levels using the natural breaks method and counted the area proportion of each susceptibility class evaluated by the RF model in different precipitation levels (Figure 6). With increasing precipitation, the percentage of high-susceptibility areas gradually increased, while that of low-susceptibility areas gradually decreased. Moreover, when precipitation was moderate, the moderate-susceptibility area proportion also reached the maximum. These results confirm that the RF model can accurately identify the relationship between precipitation and landslide susceptibility.

**Table 3.** Sufficiency analysis of the dynamic LSM results.

| Model | Date | Classification | Landslide Density (/km$^2$) | Area Proportion (%) |
|---|---|---|---|---|
| RF | 2021-06-15 | Very low | 0.000005 | 72.84 |
| | | Low | 0.000088 | 11.88 |
| | | Moderate | 0.000187 | 7.11 |
| | | High | 0.000322 | 5.14 |
| | | Very high | 0.000771 | 3.03 |
| | 2020-12-15 | Very low | 0.000006 | 75.92 |
| | | Low | 0.000127 | 12.87 |
| | | Moderate | 0.000293 | 5.98 |
| | | High | 0.000452 | 3.60 |
| | | Very high | 0.000740 | 1.63 |
| ET | 2021-06-15 | Very low | 0.000004 | 68.07 |
| | | Low | 0.000062 | 14.19 |
| | | Moderate | 0.000172 | 8.97 |
| | | High | 0.000321 | 5.93 |
| | | Very high | 0.000743 | 2.84 |
| | 2020-12-15 | Very low | 0.000006 | 71.83 |
| | | Low | 0.000098 | 14.91 |
| | | Moderate | 0.000269 | 7.97 |
| | | High | 0.000443 | 3.86 |
| | | Very high | 0.000697 | 1.43 |
| LGB | 2021-06-15 | Very low | 0.000010 | 79.74 |
| | | Low | 0.000092 | 6.09 |
| | | Moderate | 0.000152 | 4.36 |
| | | High | 0.000247 | 4.11 |
| | | Very high | 0.000640 | 5.70 |
| | 2020-12-15 | Very low | 0.000015 | 83.44 |
| | | Low | 0.000144 | 6.14 |
| | | Moderate | 0.000208 | 3.89 |
| | | High | 0.000361 | 3.15 |
| | | Very high | 0.000774 | 3.38 |

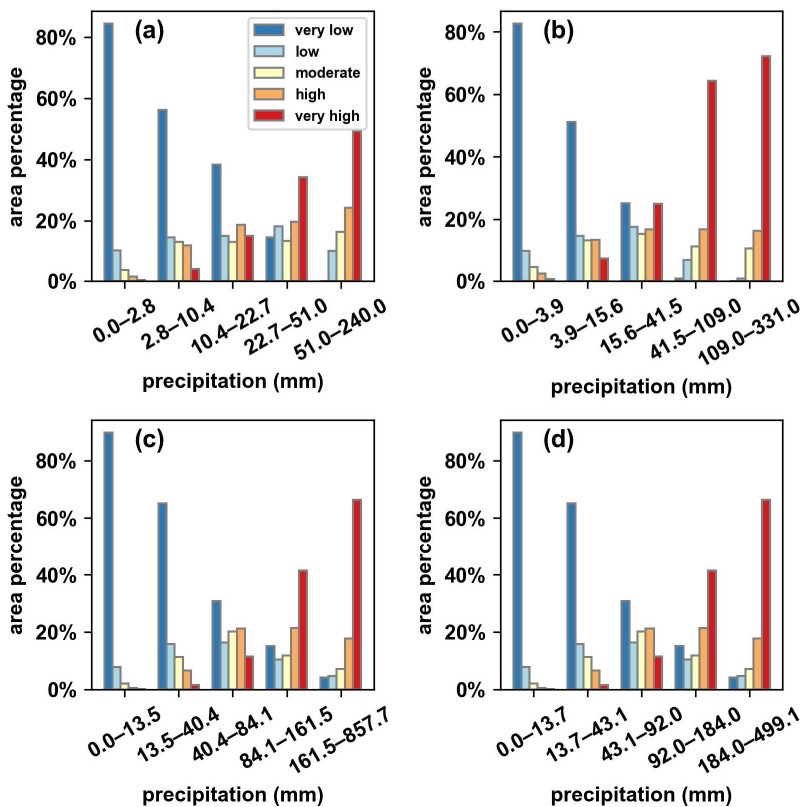

**Figure 6.** Area proportions of the five landslide susceptibility classes under different precipitation levels. Note: (**a**) Pr1d on 15 December 2020 (**b**) Pr1d on 15 June 2021 (**c**) Pr7d on 15 December 2020 (**d**) Pr7d on 15 June 2021.

*4.4. Analysis of the Cinditioning Factors*

The correlation between landslide and conditioning factors was analysed using the frequency values of landslide occurrence. For a landslide conditioning factor, frequency in a certain class is defined as the number of landslide samples divided by the total samples in this class [88]. We used the natural breaks to classify continuous conditioning factors and the result of the frequency analysis is summarised in Figure 7. For the influence of land cover type on landslide occurrence (Figure 7a), savannas, woody savannas, urban and built-up lands, and grassland were more likely to trigger landslides. This indicates that tree-poor grasslands are more prone to landslides and that human activities have strong influence on landslide occurrence. NDVI can directly reflect the vegetation cover. The frequency showed an increasing trend with the rise of NDVI, which was about 0.5 on average when the NDVI was greater than 0.33. Figure 7b–d show that the landslide frequency is high when the plan or profile curvature is negative (concave) or positive (convex), indicating that the more uneven the terrain is, the higher the landslide frequency. The number of landslides is higher on a flat surface than that on an uneven surface due to the relatively low proportion of pixels with the higher absolute value of curvature on a global scale (Figure S1i,j). Figure 7e indicates that frequency increases markedly with an increasing slope. Figure 7f,g illustrate a close relationship between both Pr1d and Pr3d and landslide occurrence. The frequency of landslides increases significantly with growing rainfall. Although extreme rainfall (pr1d reaches 25 mm or pr7d reaches 115 mm) occurs rarely, it is very likely to trigger landslides. It can be seen from Figure 7 h that almost all landslide points were located within 73 km of roads, which is reasonable because the slope stability may have deteriorated due to the change of topography and landform caused by human activities. Figure 7i reveals the relationship between the distance to fault and the landslide occurrence. When the distance to fault was 0–169 km, the number of landslide points and frequency were highest and the frequency of other distance classes was lower than

0.4; the frequency and the number of landslide points dropped rapidly when the distance to fault was between 169 km and 1156 km and rebounded when the distance was 1156–2354 km; as the distance increased further, the frequency gradually approached 0. With the increase of TWI, the number of landslides and frequency showed an increasing trend first and then a decreasing trend, and they reached the peak when TWI was between 5.47–7.29 (Figure 7j).

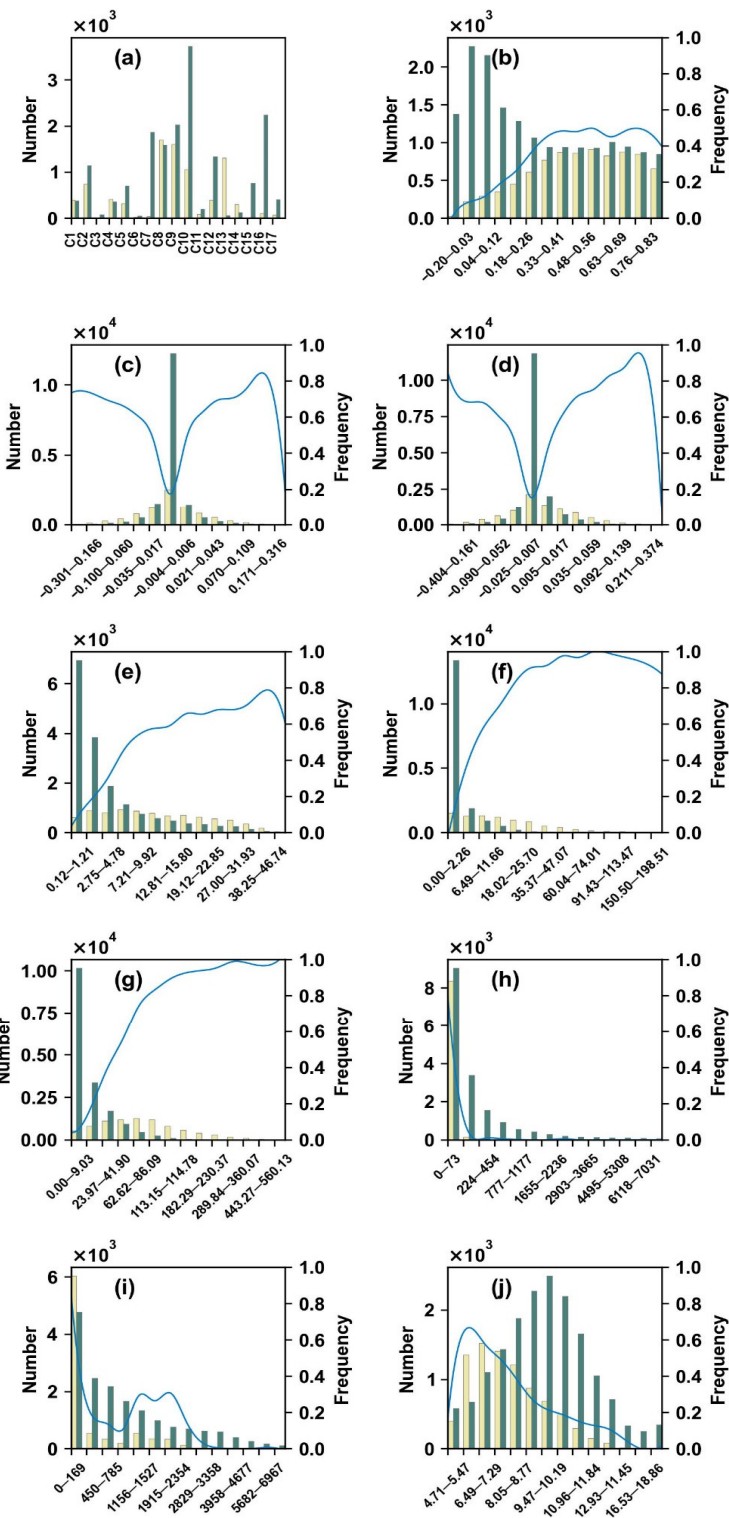

**Figure 7.** Relationship between samples and landslide conditioning factors, (**a**) land cover (**b**) NDVI (**c**) plan curvature (**d**) profile curvature (**e**) slope (°) (**f**) Pr1d (mm) (**g**) Pr7d (mm) (**h**) distance to road (km) (**i**) distance to fault (km) (**j**) TWI.

Figure 8 shows the relative importance of each landslide conditioning factor using the RF algorithm. Dynamic features, including Pr1d, Pr7d, NDVI, and land cover, accounted for 50.1% of the total information contribution, among which the precipitation features contributed 43.4% (Pr7d and Pr1d together). Pr7d, the most important landslide conditioning factor, accounted for approximately 24.4% of the total information contribution. These results indicate that dynamic features are indispensable explanatory variables for enhancing modelling accuracy. Furthermore, precipitation, environmental and topographic features are the most dominant features in modelling, collectively accounting for 94.4% of the total information contribution. In addition, the information contribution of distance to road was the second-highest, which shows that the influence of human activities on the occurrence of landslides cannot be ignored.

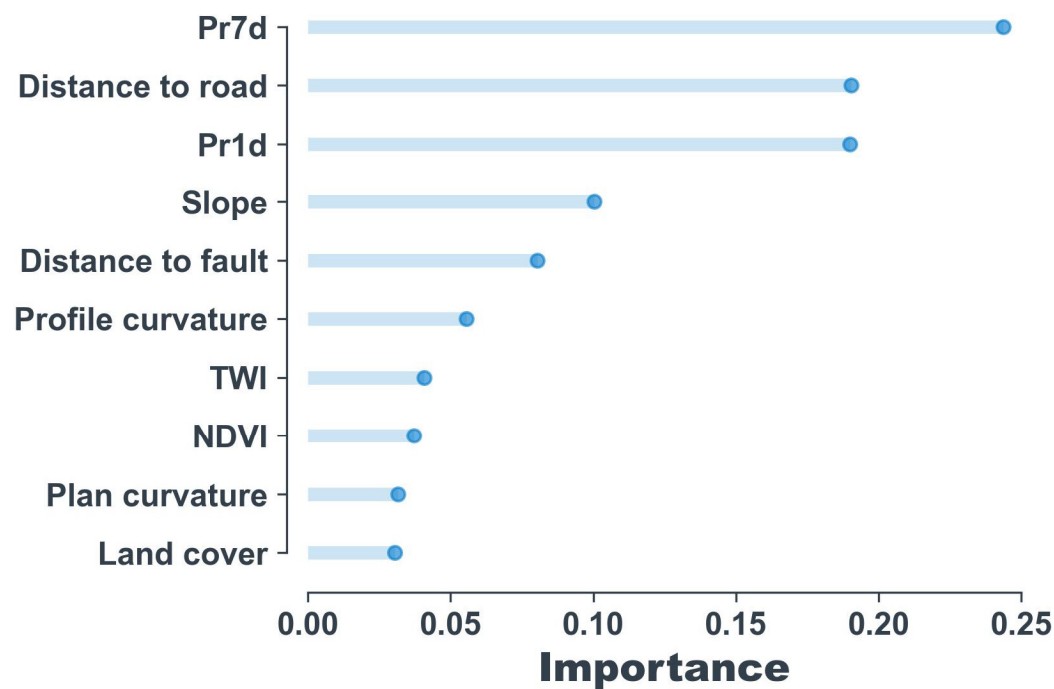

**Figure 8.** Relative importance of the conditioning factors in the global dynamic rain-fall induced LMS model for using RF.

## 5. Discussion

### 5.1. Difference from Previous Works

In order to clarify the difference between this study and other related works, we selected certain representative works on a large scale for comparison. Lin et al. [45] produced the global landslide susceptibility map using logistic regression based on the precipitation trigger factors, including total monthly rainfall and annual rainfall, as well as geological factors, topographic factors and soil factors etc. The model constructed by the above work has good performance (AUC = 0.88) and it has good applicability in the world. However, the above work is similar to most landslide susceptibility studies, which do not give a specific time scale, and which focus on describing the relative possibility of long-term landslide occurrence in the world. As for our study, the way to obtain the long-term landslide susceptibility map is to produce the landslide susceptibility map for each day in a specific period and take the average value. Such a method can obtain a landslide susceptibility map with higher model accuracy (AUC = 0.975) and a clear time scale. In addition, Kirschbaum and Stanley [43] developed a landslide hazard assessment

for situational awareness (LHASA) to indicate potential landslide activity in near real time. Based on a global landslide susceptibility map [44] and combined with the extreme precipitation threshold, LHASA shows a hierarchical nowcast of landslide hazards. We consider that not only is the landslide hazard seasonal, but the susceptibility can also reflect the seasonal pattern. Our study emphasises dynamic landslide LSM to improve the quality of landslide susceptibility map, and LHASA is divided into two parts: one is to produce static landslide susceptibility map (AUC = 0.83) and the other is to predict landslide hazard based on a decision tree. Furthermore, the model produced in this study has better performance and may be used to improve landslide hazard prediction accuracy.

*5.2. Uncertainty Analysis*

There are still many uncertainties in our global, dynamic rainfall-induced LSM model. Firstly, there is uncertainty in the spatial locations of the landslide points in COOLR, and these location errors can lead to a mismatch between the sample points and the corresponding landslide influencing factors.

Secondly, the choice of spatial resolution for the landslide influencing factors affects the modelling results. This is because the spatial variability may vary among different landslide influencing factors; precipitation, for example, is relatively consistent on a large scale, while topographic factors may show continuous variation within a small area. In addition, the proposed model was constructed based on the assumption that landslides are more likely to occur under conditions similar to those of past landslides and that static variables (such as elevation, slope, and soil features) remain constant. However, landslide susceptibility changes as the environment changes, which means that the whole process is time-dependent [26]. Thus, additional dynamic changes in landslide influencing factors and improved accuracy of landslide points for model training are expected to improve the dynamic LSM accuracy.

Finally, we selected a large number of landslide influencing factors according to the mechanisms of landslides and used machine learning algorithms that are less disturbed by the correlations between explanatory variables to construct the model; nevertheless, it remains difficult to guarantee that all landslide influencing related factors are fully considered since the selection of landslide influencing factors plays a decisive role in LSM. Fortunately, with the development of multisource remote sensing, the Global Navigation Satellite System (GNSS), and the emergence of various new time-series datasets, the identified locations of landslide events will be more accurate in the future, and landslide influencing factors can be obtained with higher spatiotemporal resolutions. Thus, the global, dynamic rainfall-induced LSM model can be further improved to more accurately capture spatial variation in landslide susceptibility.

*5.3. Application of Machine Learning Algorithm*

A comparison of the three machine learning algorithms used in this study reveals that certain algorithms (such as LGB) are not applicable to LSM, even though they may perform very well on binary classification tasks. LGB gives highly extreme classification probability assessments, and the mapping results often have difficulty in conforming to natural patterns of landslide. Nevertheless, it should be noted that this conclusion is limited to the current study area and time scale based on the landslide inventory used in this study. Therefore, the most suitable machine learning algorithm for LSM cannot be evaluated by using only metrics such as AUC and ACC. Instead, sufficiency analyses for LSM should be a critical step.

Moreover, we chose three tree-based models not only because of their high performance but also because tree-based models do not require the features to be normalised and the category variables do not need to be subjected to processing such as one-hot encoding. In general, clustering, SVM, and neural network algorithms all need to make the feature matrix dimensionless, both for modelling and predicting, but this can introduce substantial difficulties in the generation of global high-resolution dynamic susceptibility

maps. Consequently, these other approaches cannot meet the modelling, evaluation, and speed requirements of dynamic LSM.

### 5.4. Use of Dynamic LSM Model

The concept of dynamic LSM models exposes another framework for analysing hazard susceptibility, enabling the results of hazard susceptibility mapping to be applied with greater accuracy in a wide range of fields, such as assessing changes in global landslide risk based on future dynamic features. Furthermore, the same approach can be applied to mapping the landslide susceptibility with different triggers, assessing forest wildfire and flood susceptibility, and so on, which can enhance the accuracy of different hazard susceptibility modelling tasks.

The application of the model can be divided into two parts, producing a historical dynamic landslide susceptibility map and predicting global real-time landslide susceptibility. To carry out real-time prediction based on this model, it is vital to obtain real-time precipitation data. We can use GFS precipitation data to supplement GPM real-time rainfall data. GFS data provides a 384-h precipitation forecast. Both GPM and GFS data can be loaded directly from GEE. Meanwhile, other dynamic landslide conditioning factors NDVI and landcover can be easily obtained from GEE. Admittedly, the immediacy of land cover and NDVI data is not enough, but their dynamics over a short period are relatively small, which does not have a significant impact on the result. The cases in the result section show historical dynamic LSM. Here, we give an example of global landslide susceptibility prediction (Figure 9). On August 24, we used the GFS data from August 22 to 25 and GPM data from August 19 to 21, the most recent NDVI and land cover data to predict the landslide susceptibility of August 25. Relevant dynamic landslide conditioning factors are visualised in Figure S3.

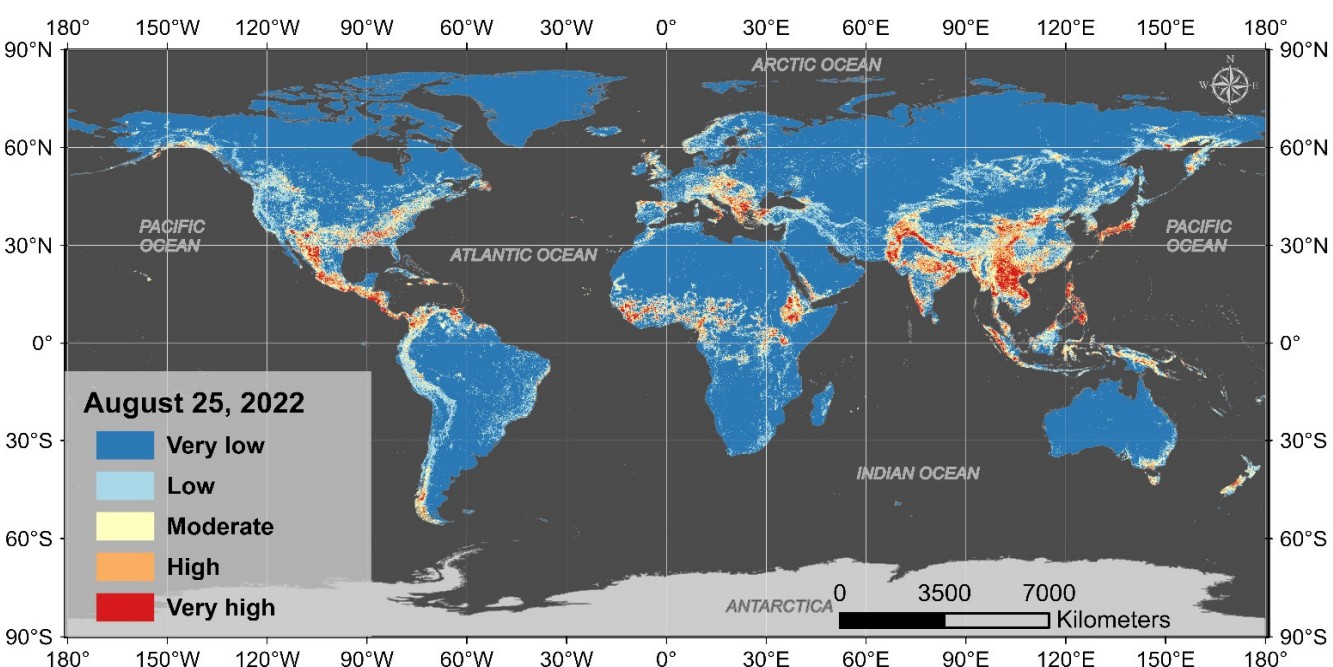

**Figure 9.** Landslide susceptibility prediction result for 25 August 2022.

### 6. Conclusions

In this study, we developed a global, dynamic rainfall-induced LSM model using the RF algorithm. GEE was adopted as the main data platform and feature extraction tool to generate a feature matrix combined with a global landslide inventory. The performance of three different machine learning algorithms (LGB, RF, and ET) on this feature matrix was then compared. We evaluated the model performance based on ACC, AUC, and other

commonly used metrics in binary classification and split the feature matrix into a training dataset and a testing dataset for modelling. We discovered that LGB, RF, and ET each yielded excellent classification results, but LGB displayed the best binary classification performance. Nevertheless, by comparing the dynamic LSM results of these models, we concluded that, although LGB performed better in binary classification, it is not applicable to global dynamic LSM. In contrast, the assessment results of the RF model are smooth, reliable, and robust, and RF can obtain satisfactory results under a variety of scenarios. We will publish this model embedded in GEE. This model can provide technological support for dynamic rainfall-induced landslide susceptibility assessment on a global scale, which can be useful for land use planning and risk analysis.

**Supplementary Materials:** The following supporting information can be downloaded at: https://www.mdpi.com/article/10.3390/rs14225795/s1, Figures S1 and S2: all the features used by the global dynamic rainfall-induced landslides susceptibility mapping model; Figure S3: the dynamic landslide influencing factors used in the prediction case; Table S1: the data sources and citations for the landslide points used [96–104]. References [96–104] are cited in the Supplementary Materials.

**Author Contributions:** Conceptualization, K.L. and M.W.; methodology, B.L., W.Z. and Z.J.; software, B.L.; validation, N.Q.; writing—original draft preparation, B.L.; writing—review and editing, B.L. and Q.H.; visualization B.L. and Q.H. All authors have read and agreed to the published version of the manuscript.

**Funding:** This work was supported by the National Natural Science Foundation of China (grant number 41771538). The financial support is highly appreciated.

**Data Availability Statement:** All data sources are described in the article. The developed dynamic landslide susceptibility mapping model can be obtained by sending an email to the corresponding author. The examples dynamic landslide susceptibility maps produced in this study can be downloaded at https://github.com/Chuiniuer/global-dynamic-landslide-susceptibility-maps/tree/master, accessed on 10 November 2022.

**Conflicts of Interest:** The authors declare that they have no known competing financial interests or personal relationships that could have appeared to influence the work reported in this paper.

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
