# Peer review of "Global Dynamic Rainfall-Induced Landslide Susceptibility Mapping Using Machine Learning"

_remotesensing, doi:10.3390/rs14225795_

Round 1

Reviewer 1 Report

This manuscript (remotesensing-1982374) tries to perform a landslide susceptibility mapping in entire global world by comparing three commonly-used machine learning models (namely RF, Random Forest, ERT, Extremely Randomized Trees, and LightGBM, Light Gradient Boosting Machine). Although it is an easy-to-follow manuscript, it is not entirely new to use these common machine learning models in geohazard susceptibility mapping. Another very serious concern is that some related studies have been neglected. A further and detailed literature review must be conducted. Also, the current results of this study can hardly be reviewed because of those problems on data and methodology. Therefore, at least a “Major Revision” is required. My suggestions and comments are presented as follows:

- 1. The scientific question is missing in the Abstract. Similarly, the Introduction Section is not strong because the authors failed to raise an important scientific question. Therefore, potential readers can hardly identify the need that the authors should have to provide a new solution from an international perspective. What I have learned from the introduction is that the authors applied several previous established models to various different dates (i.e., the so-called "dynamic"). Note that Random Forest, Extremely Randomized Trees, and Light Gradient Boosting Machine are not new methods in landslide and/or geohazard susceptibility mapping.

-2. In Line 60, the authors mentioned that: "no specific model has been shown to be suitable for all LSM scenarios". The same question for the analysis in this study. The result was merely subject to the study area and study period.

-3. In Line 92~94, the authors mentioned that: "However, when constructing a landslide susceptibility model at the global scale, dynamic LSM is burdened by an overabundance of data regarding the dynamic factors of all landslide events". Nevertheless, the authors need to answer clearly why we must build the landslide susceptibility model on the global scale? Actually, the influencing factors will be very different across different countries and regions because of the existence of significant spatial heterogeneity.

-4. Another very serious concern is that the authors must look further into the latest research in related field. In fact, the literature review is far from enough. In particular, the maximum entropy (MAXENT) algorithm has been successfully used in geohazard susceptibility mapping. However, this well-accepted technique is totally ignored in the manuscript, and the following articles should be cited.

Predicting future urban waterlogging-prone areas by coupling the maximum entropy and FLUS model. Sustainable Cities and Society, 2022, 80: 103812.

Assessing fire hazard potential and its main drivers in Mazandaran province, Iran: a data-driven approach. Environ Monitoring Assess 190, 670 (2018).

Land subsidence hazard modeling: Machine learning to identify predictors and the role of human activities. Journal of Environmental Management, 2019, 236:466-480.

-5. The authors have extracted only the landslide events that were triggered by precipitation between 2000 and 2020. Firstly, please explain clearly why this specific period was selected. Secondly, are the landslide events observed in more than twenty years ago still active? It should be much better to remain only those frequently-happened landslide points. Besides, did the authors utilize the long-term multi-temporal spatial data given that the landslide data are from 2000~2020?

-6. Line 123~128: The authors need to explain clearly why and why only those 17 influencing factors about landslide susceptibility mapping were selected.

-7. In Materials Section: the authors failed to provide the specific details of some input datasets, such as the dates in acquiring them, and the accuracies of the data. I suggest the authors to list all the information in Table 1. Besides, why the authors only used the Global Elevation Data in 2010?

-8. The Global Seismic Hazard Map data only have a spatial resolution of 0.4 degree, and the ERA5-Land Hourly – ECMWF Climate Reanalysis data only have a resolution of 0.1 degree, which are all too coarse for detailed analysis (~0.005 degree or ~500 m in this study). These uncertainties will affect the reliability of the modeling results.

-9. The authors need to explain clearly why these three machine learning methods were selected? For example, why not use the more common MAXENT and artificial neural network models?

-10. Figure 3 and its explanations should be moved to the Results Section 4.

-11. Line 231~233: The authors need to explain clearly why this research set the positive and negative sample ratio to 1:2. In fact, there is no best splitting ratio for geohazard susceptibility mapping. The most suitable splitting ratio will vary with the study cases.

-12. Line 238: The authors need to explain clearly why the aspect was considered? For example, which aspects are more easily affected by landslides? And why?

-13. The determination of the parameters of these three machine learning models should be clearly explained.

-14. Line 305~308: the landslide susceptibility probability map was divided into five categories by using the equal interval classification method with class names (probability ranges) of very low (0–0.2), low (0.2–0.4), moderate (0.4–0.6), high (0.6–0.8) and very high (0.8–1.0). What are the criteria? In fact, these criteria should be defined separately according to different models.

-15. I can hardly identify the differences among the different maps for those three models in Figure 5. Please improve this figure.

-16. The accuracy values for both training and testing samples should be presented. The current results of this study can hardly be reviewed because of those problems about data and methodology.

-17. The standard of English is low in places, such as "Error! Reference source not found" in many sentences. The paper should be proofread by a native English speaker.

Author Response

Dear Reviewer,

We have responsed to every question. Please see the attachment.

Kind regards,

Kai Liu, on behalf of all co-authors.

Reviewer 2 Report

Review comments on the manuscript

“Global dynamic rainfall-induced landslide susceptibility mapping using machine learning”

Bohao Li , Kai Liu, Ming Wang , Qian He , Ziyu Jiang , Weihua Zhu and Ningning Qiao

1.  What are the criteria to select the causative factors responsible for landslides all over the world? Do the authors agree with the fact that, these 17 factors fit for the whole world irrespective of the geology, geomorphology, triggering factors, type of materials etc.?

2. What is the role of earthquake (PGA) in rainfall induced landslide susceptibility mapping?

3. Can it be possible for a reader or researcher to determine the landslide susceptibility of their country from these maps?

4. A total of 9223 rainfall induced landslides during the period of 2000-2020 has been considered. What is the credibility of this fact? Authors are suggested to give a review on the LSM generated for whole world along with the number of landslides considered in the landslide inventory.

Author Response

(The authors gave the same response as above.)

Reviewer 3 Report

    The authors give a GLOBAL landslide susceptibility mapping using machine learning in this manuscript. After reviewing the entire article, the reviewer thinks that the structure, logic and language basically meet the standards for publishing. However, the most deficiency is still the general one of this type of study, not bringing forth new ideas.

  Generally, this manuscript uses the regular machine learning methods  and models inside. The difference is just the study region expanding into the earth-sized, with higher accuracy in AUC. I fail to see the innovation in this manuscript, in spite of all the fine figures presented in the supplimentary file.

  I suggest a Major Revision. Publication is going to be considered if the necessary content the really new ideas or improvement are given in the next revision.

Author Response

(The authors gave the same response as above.)

Round 2

Reviewer 1 Report

Thank you for incorporating my previous comments and suggestions.

Reviewer 3 Report

  Generally, the authors have made efforts to improve the manuscript. After the revison by the authors, the weak description of novelty in the first manuscript has been fixed. The latest manuscript has met the standard for publication.